# Structural Diversity of Mercury(II) Halide Complexes Containing Bis-pyridyl-bis-amide with Bulky and Angular Backbones: Ligand Effect and Metal Sensing

**DOI:** 10.3390/ijms23147861

**Published:** 2022-07-16

**Authors:** Manivannan Govindaraj, Wei-Chun Huang, Chia-Yi Lee, Venkatesan Lakshmanan, Yu-Hsiang Liu, Pamela Berilyn So, Chia-Her Lin, Jhy-Der Chen

**Affiliations:** 1Department of Chemistry, Chung-Yuan Christian University, Chung Li, Taoyuan 320, Taiwan; manivannanjent@gmail.com (M.G.); learn122568@gmail.com (W.-C.H.); miss10031031@gmail.com (C.-Y.L.); flower95@gmail.com (V.L.); g10963021@cycu.edu.tw (Y.-H.L.); 2Department of Chemistry, National Taiwan Normal University, Taipei 106, Taiwan; pbtiuso@gmail.com

**Keywords:** Hg(II) complex, Hg(II) coordination polymer, crystal structure analysis, halide anion effect

## Abstract

Hg(II) halide complexes [HgCl_2_] 2**L^1^** [**L^1^** = *N,N’*-bis(3-pyridyl)bicyclo(2,2,2,)oct-7-ene-2,3,5,6-tetracarboxylic diamide), **1**, [HgBr_2_(**L^1^**)]_n_, **2**, [HgI_2_(**L^1^**)], **3**, [Hg_2_X_4_(**L^2^**)_2_] [X = Cl, **4**, Br, **5**, and I, **6**; **L^2^** = *N,N’*-bis(4-pyridylmethyl)bicyclo(2,2,2,)oct-7-ene-2,3,5,6-tetracarboxylic diamide] and {[HgX_2_(**L^3^**)]⋅H_2_O}_n_ [X = Cl, **7**, Br, **8** and I, **9**; **L^3^** = 4,4′-oxybis(N-(pyridine-3-yl)benzamide)] are reported and structurally characterized using single-crystal X-ray diffraction analyses. The linear HgCl_2_ units of complex **1** are interlinked by the **L^1^** ligands through Hg---N and Hg---O interactions, resulting in 1D supramolecular chains. Complex **2** shows 1D zigzag chains interlinked through the Br---Br interactions to form 1D looped supramolecular chains, while the mononuclear [HgI_2_**L^2^**] molecules of **3** are interlinked through Hg---O and I---I interactions, forming 2D supramolecular layers. Complexes **4**–**6** are isomorphous dinuclear metallocycles, and **7**–**9** form isomorphous 1D zigzag chains. The roles of the ligand type and the halide anion in determining the structural diversity of **1**–**9** is discussed and the luminescent properties of **7**–**9** evaluated. Complexes **7**–**9** manifest stability in aqueous environments. Moreover, complexes **7** and **8** show good sensing towards Fe^3+^ ions with low detection limits and good reusability up to five cycles, revealing that the Hg-X---Fe^3+^ (X = Cl and Br) interaction may have an important role in determining the quenching effect of **7** and **8**.

## 1. Introduction

The investigation of the rational design and synthesis of novel coordination polymers (CPs) continues to be an intense area of research due to their interesting structural diversity and potential industrial applications [1,2,3]. Although many remarkable CPs have been reported, it remains elusive to predict the structural types of the various CPs prepared. While choosing the appropriate metal cations and organic spacers is essential, the structural diversity of CPs is also subject to the identity of the counterions and the reaction conditions involved, such as the metal-to-ligand ratio, the solvent system, and the reaction temperature. In the same way, the halide anions have shown significant influence on the structures of the Hg(II) complexes, but it is difficult to predict which anions give similar or different structures [4,5,6,7,8,9,10]. In some cases, the chloride and bromide anions have no contribution on structural diversity, but the iodide anion has; while in other cases, the halide anions yield the same contribution to the structures [11,12,13]. For example, the reactions of HgX_2_ (X = Cl, Br and I) with the ligand *N,N,N′,N′*-tetraisopropyl-3,4-pyridinedicarboxamide afforded 2D CPs for the three anions, whereas those with *N,N,N′,N′*-tetraisobutyl-3,4-pyridinedicarboxamide gave 2D CPs for the chloride and bromide anions and a dimeric complex for the iodide anion [6]. On the other hand, the complexes formed by the reactions of HgX_2_ with 2-pyridine piconyl hydrazone, 2-acetylpyridine piconyl hydrazone, or 2-phenylpyridine piconyl hydrazone are all mononuclear, showing interesting weak interactions that differentiate these complexes [7]. However, using the bis-(3-pyridyl)isophthalamide) ligand, bimetallic macrocycles for X = Cl and Br and 1D CP for X = I were produced, respectively [11]. The effect of the halide anion on the structural diversity of the formamidinate-based CPs has also been reported. While the 3D (X = Cl) and 2D (X = Br and I) heteronuclear CPs based on quadruple-bonded dimolybdenum units were obtained from the reactions of [Mo_2_(4-pyf)_4_] (4-Hpyf = 4-pyridylformamidine) with HgX_2_ [12], the reactions of 4-Hpyf with HgX_2_ afforded a 2D layer for X = Cl and 1D helical chains for X = Br and I, respectively [13].

Previously, we reported several bis-pyridyl-bis-amide (bpba)-based 1D Hg(II) halide CPs [14,15,16,17]. By using the rigid and isomeric 2,2-(1,2-phenylene)-bis(N-pyridin-3-yl)acetamide (1,2-pbpa), 2,2’-(1,3-phenylene)-bis(N-(pyridin-3-yl)acetamide (1,3-pbpa), and 2,2-(1,4-phenylene)-bis(N-(pyridin-3-yl)acetamide (1,4-pbpa), several CPs showing 1D zigzag, helical, mesohelical, and sinusoidal structures were prepared [14,15,16], while using the rigid *N,N’*-di(pyridin-3-yl)naphthalene-1,4-dicarboxamide (dpndc) afforded isostructural and helical CPs [17]. The structural diversity of these Hg(II) halide CPs containing rigid bpba are thus subject to the ligand types, whereas the role of the halide anions is only suggestive [14,15,16,17]. On the other hand, the pairs of supramolecular isomers with the flexible *N,N’*-di(3-pyridyl)adipoamide (L), [HgBr_2_(GAG-L)]_n_ and [HgBr_2_(AAA-L)]_n_ and [HgI_2_(GAG-L)]_n_ and [HgI_2_(AAA-L)]_n_, exhibit mesohelical, helical, sinusoidal, and helical chains, respectively [18].

Although there are already several organic ligand-supported Hg(II) halide complexes reported, research towards understanding the general effect of the halide anion on their structural diversity is less. Moreover, it is well known that the bpba ligands may easily be tailored to form structures with different flexibility and different shapes [14,15,16,17,18]; thus, in this report, we intend to investigate the effect of the halide anion on the structural types of the Hg(II) halide complexes containing bpba ligands with a bulky and angular backbone. We prepared two bpba ligands with bulky backbones, namely *N,N’*-bis(3-pyridyl)bicyclo(2,2,2,)oct-7-ene-2,3,5,6-tetracarboxylic diamide (**L^1^**), Figure 1a, and *N,N’*-bis(4-pyridylmethyl)bicyclo(2,2,2,)oct-7-ene-2,3,5,6-tetracarboxylic diamide (**L^2^**), Figure 1b, as well as a bpba ligand with an angular backbone, namely 4,4′-oxybis(N-(pyridine-3-yl)benzamide) (**L^3^**), Figure 1c. Their corresponding reactions with Hg(II) halide salts were carried out. The synthesis and structures of [HgCl_2_]⋅2**L^1^**, **1**, [HgBr_2_**L^1^**]_n_, **2**, [HgI_2_**L^1^**], **3**, [Hg_2_X_4_(**L^2^**)_2_] (X = Cl, **4**; Br, **5**; I, **6**), and [HgX_2_(**L^3^**) H_2_O]_n_ (X = Cl, **7**, Br, **8** and I, **9**) form the subject of this report, and the effect of the ligand type and halide anion on the structural diversity is discussed. The luminescence properties of **7** and **8** provide a unique opportunity to investigate the role of the halide anions in determining the sensing properties of the 1D bpba-based Hg(II) CPs.

## 2. Results and Discussion

### 2.1. Crystal Structure of ***1***

Crystals of **1** conform to triclinic space group *P*ī with a half Hg(II) ion, one chloride anion, and one **L^1^** ligand in the asymmetric unit. Figure 2a shows the coordination environment of the Hg(II) metal center, which is two-coordinated by two symmetry-related chloride anions [Hg-Cl = Hg-Cl(A) = 2.3055(5) Å], resulting in a linear geometry for the Hg(II) ion [∠Cl-Hg-Cl(A) = 180°]. Moreover, the linear metal units interact with the **L^1^** ligands in two directions, orthogonal to the linear metal unit through Hg---N [Hg---N(1) = Hg---N(1A) = 2.727(1) Å] and Hg---O [Hg---O(3B) = Hg---O(3C) = 3.122(1) Å] interactions, resulting in octahedral fashions for the Hg(II) ions, leading to the formation of a 1D linear supramolecular chain, Figure 2b. The sum of the van der Waals radius of Hg and N is 3.07 Å, and that of Hg and O is 3.10 Å (van der Waals radius: Hg = 1.55, N = 1.52, and O = 1.55 Å).

### 2.2. Crystal Structure of ***2***

Crystals of **2** conform to triclinic space group *P*ī with one Hg(II) ion, two bromide anions, and one **L^1^** ligand in each asymmetric unit. Figure 3a shows the coordination environment of the Hg(II) metal center, which is four-coordinated by two nitrogen atoms from two **L^1^** ligands [Hg-N(1) = 2.373(3); Hg-N(4A) = 2.435(3) Å] and two Br^−^ anions [Hg-Br(1) = 2.4472(5); Hg-Br(2) = 2.5180(4) Å], resulting in a distorted tetrahedral geometry (τ_4_ = 0.81) for the Hg(II) ion, with bond angles of N(1)-Hg-N(4A) = 89.77(11), N(1)-Hg-Br(1) = 110.48(8), N(4A)-Hg-Br(1) = 104.32(8), N(1)-Hg-Br(2) = 102.94(7), N(4A)-Hg-Br(2) = 105.28(8) and Br(1)-Hg-Br(2) = 134.87(2). The Hg(II) cations are connected by the **L^1^** ligands to form 1D zigzag chains, which are further linked by the bromide anions through the Br---Br interactions of 3.646(1) Å, that is significantly shorter than the sum of two van der Waals radii of Br (3.70 Å), resulting in 1D looped supramolecular chains, Figure 3b.

### 2.3. Crystal Structure of ***3***

Crystals of **3** conform to triclinic space group *P*ī with one Hg(II) ion, two iodide anions, and one **L^1^** ligand in each asymmetric unit. Figure 4a shows the coordination environment of the Hg(II) metal center of the mononuclear **3**, which is three-coordinated by one nitrogen atom [Hg-N(1) = 2.362(3) Å] and two iodide anions [Hg-I(1) = 2.6272(3) Å; Hg-I(2) = 2.6290(3) Å], resulting in a trigonal planar geometry for the Hg(II) ion [∠I(1)-Hg-I(2) = 151.699(8)°; ∠N(1)-Hg-I(1) = 101.72(6)°; ∠N(1)-Hg-I(2) = 106.19(6)°]. Moreover, the molecules of **3** are interlinked through Hg---O [2.949(3) and 2.844(3) Å] and I---I [3.8791(4) Å] interactions, leading to the formation of a 2D supramolecular layer, Figure 4b. The I---I distance of 3.8791(4) Å is significantly shorter than the sum of two van der Waals radii of the iodo atom, which is 3.96 Å.

### 2.4. Crystal Structures of ***4***–***6***

Complexes **4**–**6** are isomorphous, and their crystals conform to monoclinic space group *C*2*/c* with one Hg(II) cation, two halide anions, and one **L^2^** ligand in the asymmetric unit. Figure 5 shows a representative drawing for the dinuclear structures of **4**–**6** (X = Cl, **4**; Br, **5**; I, **6**). The coordination environment of the Hg(II) metal center is four-coordinated by two nitrogen atoms from two **L^2^** ligands and two halide anions, resulting in a distorted tetrahedral geometry (τ_4_ = 0.71 for **1**; 0.73 for **2**; 0.75 for **3**). Table 1 lists the selected bond distances and angles for **4**–**6**. The dihedral angles between the two phenyl rings of **L^2^** are 59.24, 61.25, and 63.300, respectively. While the Hg-X distances and the dihedral angles increase from **4** to **6**, the X-Hg-X angles decrease, showing the size effect of the halide anions. Moreover, the molecules of the dinuclear complexes **4**–**6** are linked by the intermolecular C-H---O (H---O = 2.509–2.539 Å, ∠C-H---O = 113.9–143.8° for **4**; H---O = 2.522–2.587 Å, ∠C-H---O = 114.2–137.0° for **5**; H---O = 2.532–2.638 Å, ∠C-H---O = 113.9–132.7° for **6**), and **5**–**6** are also linked by the intermolecular C-H---X (H---Br = 3.069, 3.115 Å, ∠C-H---Br = 130.3, 118.5°; H---I = 3.186, 3.281 Å, ∠C-H---I = 132.5, 123.2°) interactions, resulting in 3D supramolecular structures, Appendix A.

### 2.5. Crystal Structures of ***7***–***9***

Single-crystal X-ray diffraction analysis reveals that the isomorphous complexes **7**–**9** are crystallized in the monoclinic space group *P*2_1_/*c* with one Hg(II) ion, one **L^3^** ligand, two halide anions, and one co-crystallized water molecule in each asymmetric unit. Figure 6a depicts a representative drawing showing the coordination environment of the Hg(II) centers. The central Hg (II) ions are four-coordinated by two pyridyl nitrogen atoms from two **L^3^** ligands and two halide anions, resulting in tetrahedral geometries, which are further linked by the **L^3^** ligands to form 1D zigzag chains, Figure 6b. Table 2 lists the selected bond distances and angles for **7**–**9**. While the Hg-N distances are similar, the Hg-X distances increase from Cl to I and the X-Hg-X angles decrease. Moreover, the zigzag chains of **7**–**9** are linked by the intermolecular C-H---O (H---O = 2.392, 2.479 Å, ∠C-H---O = 136.3, 141.6° for **7**; H---O = 2.395, 2.522 Å, ∠C-H---O = 140.5, 142.8° for **8**; H---O = 2.456, 2.599 Å, ∠C-H---O = 143.3, 147.9° for **9**) and the intermolecular C-H---X (H---Cl = 2.851, 2.878 Å, ∠C-H---Cl = 150.7, 131.6°; H---Br = 2.916, 2.979 Å, ∠C-H---Br = 158.2, 132.9°; H---I = 3.065, 3.154 Å, ∠C-H---I = 143.3, 133.0°) interactions, as well as the N-H---O (H---O = 1.990 Å, ∠N-H---O = 161.1° for **7**; H---O = 2.059 Å, ∠N-H---O = 159.4° for **8**; H---O = 2.160 Å, ∠N-H---O = 156.7° for **9**) interactions with the amide oxygen atoms, resulting in 3D supramolecular structures, Appendix A.

Some additional parameters were manipulated to probe the structural differences in the isomorphous **7**–**9**, where distances d1 and d2 are distances from the bridging oxygen atom to the two Hg(II) ions and d3 is the distance between the two Hg(II) ions bridged by the **L** ligand, and angles θ1, θ2, and θ3 are the dihedral angles differences (Figure 7 and Table 3). The dihedral angles θ2 and θ3 increased from **1** to **3**, while the θ1 values are in a reverse order, indicating the effect of the size of the halide anion on the structures.

### 2.6. Effect of Halide Anion and Ligand Type on Structural Diversity

The structural types of complexes **1**–**9** are listed in Table 4. While complexes **1**–**3** containing rigid-bulky **L^1^** display different structures that are subject to the nature of the halide anion, the structural diversity of the flexible-bulky **L^2^**-based **4**–**6** and the angular **L^3^**-based **7**–**9** is independent of the nature of the halide anion. The halide anions play different structure-determining roles in **1**–**9** containing different bpba ligands. The halide anion effect on the structural types of the Hg(II) CPs is thus subject to the identity of the bpba ligand.

### 2.7. Luminescence Properties

Complexes **7**–**9** that contain d^10^ Hg(II) ions and organic ligand **L^3^** with a large π-conjugated system may exhibit fluorescent properties [19]. Therefore, their solid-state emission spectra were examined at room temperature, as shown in Figure 8 and Table 5. The free **L^3^** ligand shows an emission at 430 nm upon excitation at 330 nm, which could be due to the n → π* or π → π* transitions, while complexes **7**–**9** exhibit emission bands at 420, 416, and 400 nm, respectively. Due to the d^10^ electronic configuration of the Hg(II) metal ion that hardly undergoes either oxidation or reduction, the emissions of **7**–**9** may thus result from the organic linkers and are attributable to n → π* or π → π* transitions [20]. Noticeably, the emission wavelengths of **7**–**9** are similar with decreasing intensities from Cl, Br, to I, indicating the heavy atom effect of the halide anions [21]. It is noted that no detectable emission can be found for the **L^1^**, **L^2^** and their complexes **1**–**6**, which can be ascribed to the different natures of **L^1^**^,^
**L^2^**, and **L^3^**.

### 2.8. Mechanochemical Synthesis and Stability of Complexes ***7***–***9***

To obtain complexes **7**–**9** efficiently, we studied their mechanochemical synthesis. Manual grinding of mercury(II) halide salts with **L^3^** in methanol/H_2_O or ethanol/H_2_O afforded complexes **7**–**9**, which were verified by characterization using PXRD. As shown in Appendix A, the PXRD patterns of the samples prepared using the solvothermal reactions and mechanochemical reactions matched quite well, indicating the bulk purities of **7**–**9**. The PXRD patterns of the mechanochemical products were comparatively broad, most probably due to the fact that mechanochemical products generally have lower crystallinity compared with the solvothermal product. Additionally, the mechanochemical method was only successful in the MeOH/H_2_O and EtOH/H_2_O solvent systems, while in various solvents such as pure H_2_O, CH_2_Cl_2_, and MeCN, different products were obtained, indicating the solvent selectivity of complexes **7**–**9** (Appendix A). The stability of complexes **7**–**9** was studied by immersing them into H_2_O, EtOH, MeOH, CH_2_Cl_2_, and MeCN, respectively, for up to 7 days. Complexes **7**–**9** were then filtered and dried under vacuum and their PXRD patterns were measured. Appendix A show that the experimental PXRD patterns matched well with the simulated ones, indicating complexes **7**–**9** are stable in these solvents.

### 2.9. Halide Anion Effect on Metal Sensing

Complexes **7** and **8** provide a unique opportunity to investigate the effect of the halide anion on metal sensing. For the investigations, 25 mg samples of **7** and **8**, respectively, was immersed into 10 mL aqueous solutions of nitrate salts M(NO_3_)_x_ or acetate salts M(OAc)_x_ (M = Ag^+^, Cd^2+^, Co^2+^, Mn^2+^, Ni^2+^, Mg^2+^, Cu^2+^, and Fe^3+^). After 10 h, the solids were filtered and their emission spectra were measured. As shown in Appendix A, remarkable luminescence quenching of about 91% for **7** and 90% for **8** were found in the detection of Fe^3+^ ions. To further explore the quenching effect of Fe^3+^ ions, the sensing dependence of luminescence intensity on the concentration of Fe^3+^ was investigated by immersing finely ground samples (25 mg) of **7** or **8** into Fe^3+^ aqueous solutions with various concentrations (0.1 mM–1 mM) for 10 h. Appendix A show that the emission intensities were getting lower and almost completely quenched upon increasing the concentration of Fe^3+^. Quantitatively, the quenching capacity of the Fe^3+^ ion can be rationalized by the Stern–Volmer equation: I_0_/I = 1 + K_sv_ × [Q], where [Q] is the concentration of Fe^3+^, K_sv_ is the quenching constant, and I_0_ and I are the emission intensities in the absence and presence of Fe^3+^, respectively [22]. As demonstrated in Figure 9, the titration curves for Fe^3+^ ions in **7** and **8** are virtually linear at low concentrations, which gave the linear correlation coefficient (R^2^) of 0.9714 for **7** and 0.9525 for **8**, respectively, while the S-V curves at higher concentrations became nonlinear, affording Stern−Volmer constants (K_sv_) of 2.48 × 10^4^ M^−1^ for **7** and 1.2 × 10^4^ M^−1^ for **8**, respectively.

Furthermore, the detection limits were calculated according to the standard equation 3σ/k, where σ is the standard deviation from the blank measurements and k is the absolute value of the calibration curve at a lower concentration [23], giving 7.38 and 24 μM for **7** and **8**, respectively. The recyclability test showed no significant changes in the PXRD patterns (Figure 10 and Figure 11), and the luminescence intensities (Appendix A) for five regeneration cycles were consistent, indicating the reusability of **7** and **8** as sensing materials toward Fe^3+^. This demonstrates that the luminescence quenching is not due to the framework collapse of **7** and **8,** but upon the interactions with the Fe^3+^ ions. The use of Hg(II) CPs for sensing is rarely seen, and a 1D double-chain {[Hg(L)_2_]·(ClO_4_)_2_}_n_ (L = 1,3,5-tris(benzimidazolylmethyl)benzene) has been reported to show multistimuli-responsive photoluminescence sensing properties toward anions, solvents, and nitroaromatic compounds [24].

Several mechanisms for luminescence quenching such as framework collapse, cation exchange, and interactions between the incoming metal ion and the organic linker that result in competitive absorption of the excitation energies have been suggested [25]. Since the quenching of the luminescence is not due to the framework collapse, the interactions between Fe^3+^ ions and complexes **7** and **8** are the main reasons leading to the luminescence quenching [26]. The UV-Vis absorption spectrum of Fe^3+^ in aqueous solution and the corresponding excitation and emission spectra of complexes **7** and **8** are shown in Appendix A, respectively. Partial overlaps between the absorption spectrum of the Fe^3+^ ion and the excitation spectra of complexes **7** and **8** are observed, indicating that the excitation energies of **7** and **8** can be partially absorbed by the Fe^3+^ ions, and the luminescence quenching can most probably be ascribed to competitive energy absorption [25].

Moreover, the K_sv_ values, 2.48 × 10^4^ M^−1^ for **7** and 1.2 × 10^4^ M^−1^ for **8**, may indicate that the Fe^3+^ ion shows a better quenching effect to the chloride complex **7**. In addition to the possible interactions between the metal ions and the amide carbonyl oxygen atoms of the **L^3^** ligands [26], the halide anions of **7** and **8** may play an important role in determining the quenching effect. It is well known that the atomic radius of Cl^−^ is shorter than that of Br^−^, and comparatively, Cl^−^ can be regarded as a harder Lewis base than Br^−^. Since Fe^3+^ is a hard Lewis acid and interacts stronger with the Cl^−^ anion, the larger quenching effect to **7** is attributable to the formation of the stronger Hg-Cl---Fe^3+^ interaction upon the addition of the Fe^3+^ ion to complex **7**. The different quenching effect exerted by the Fe^3+^ ion may thus be ascribed to the different Hg-X---Fe^3+^ (X = Cl and Br) interactions. Energy dispersive X-ray (EDX) analysis of complexes **7**–**8** was performed after Fe^3+^ sensing (Appendix A), confirming the Fe^3+^ uptake of **7**–**8**.

## 3. Materials and Methods

### 3.1. General Procedures

Elemental analyses of (C, H, N) were performed on a PE 2400 series II CHNS/O (PerkinElmer Instruments, Shelton, CT, USA) or an Elementar Vario EL-III analyzer (Elementar Analysensysteme GmbH, Hanau, Germany). Infrared spectra were obtained from a JASCO FT/IR-460 plus spectrometer with pressed KBr pellets (JASCO, Easton, MD, USA). Powder X-ray diffraction patterns were carried out with a Bruker D8-Focus Bragg–Brentano X-ray powder diffractometer equipped with a CuKα (λ_α_ = 1.54178 Å) sealed tube (Bruker Corporation, Karlsruhe, Germany). The UV-Vis spectrum was performed on a UV-2450 spectrophotometer (Dongguan Hongcheng Optical Products Co., Dongguan, China). Emission spectra were determined with a Hitachi F-4500 fluorescence spectrophotometer (Hitachi, Tokyo, Japan). Energy dispersive X-ray (EDX) analysis was performed by using a JEOL JSM-7600F Ultra-High Resolution Schottky Field Emission Scanning Electron Microscope with Oxford Xmax80 energy dispersive X-ray spectrometer (JEOL, Ltd., Tokyo, Japan).

### 3.2. Materials

The reagents HgCl_2_ and HgBr_2_ were purchased from Acros Organics (Themo Fisher Scientific, NJ, USA) and HgI_2_ from Aldrich Chemistry Co. (Milwaukee, WI, USA). The solvents CH_3_OH (99.5%) and CH_3_CH_2_OH (99.5%) were purchased from Echo Chemical Co., Ltd. (Toufen, Miaoli, Taiwan). The **L^1^**, **L^2^**, and **L^3^** ligands were prepared according to published procedures with slight modification [27,28,29].

### 3.3. Preparations

#### 3.3.1. [HgCl_2_]⋅2**L^1^**, **1**

A mixture of HgCl_2_ (0.027 g, 0.10 mmol), **L^1^** (0.040 g, 0.10 mmol), and 10 mL EtOH was sealed in a 23 mL Teflon-lined stainless steel autoclave, which was heated under autogenous pressure to 120 °C for two days, and then, the reaction system was cooled to room temperature at a rate of 2 °C per hour. The colorless crystals suitable for single-crystal X-ray diffraction were obtained. Yield: 0.043 g (40%). Anal. Calcd for C_44_H_32_Cl_2_HgN_8_O_8_ (*M*_W_ = 1072.26): C, 49.29; H, 3.01; N, 10.45%. Found: C, 48.89; H, 2.88; N, 10.33%. FT-IR (cm^−1^): 3457 (s), 2363 (w), 2342 (w), 1715 (s), 1639 (m), 1566 (m), 1482 (m), 1429 (m), 1384 (s), 1178 (m), 1098 (w), 1048 (w), 1028 (w), 779 (m), 702 (m), 680 (m), 628 (m).

#### 3.3.2. [HgBr_2_(**L^1^**)]_n_, **2**

Complex **2** was prepared by using similar procedures for **1**, except that HgBr_2_ (0.036 g, 0.10 mmol), **L^1^** (0.040 g, 0.10 mmol), and 10 mL MeOH were used. Colorless crystals were obtained. Yield: 0.073 g (96%). Anal. Calcd for C_22_H_16_Br_2_HgN_4_O_4_ (*M*_W_ = 760.78): C, 34.74; H, 2.11; N,7.37%. Found: C, 34.75; H, 1.98; N, 7.37%. IR (cm^−1^): 3588 (m), 2361 (m), 2340 (m), 1708 (s), 1484 (m), 1434 (m), 1374 (m), 1316 (w), 1232 (w), 1200 (m), 1080 (w), 1026 (w).

#### 3.3.3. [HgI_2_(**L^1^**)], **3**

Complex **3** was prepared by using similar procedures for **1**, except that HgI_2_ (0.045 g, 0.10 mmol), **L^1^** (0.040 g, 0.01 mmol), and 10 mL MeOH were used. Colorless crystals were obtained. Yield: 0.044 g (51%). Anal. Calcd for C_22_H_16_HgI_2_N_4_O_4_ (*M*_W_ = 854.78): C, 30.91; H, 1.89; N, 6.55%. Found: C, 30.76; H, 1.84; N, 6.49%. FT-IR (cm^−1^): 3442 (s), 1722 (m), 1691 (m), 1638 (m), 1483 (w), 1430 (w), 1373 (m), 1201 (m), 1169 (w), 1050 (m), 776 (w), 729 (w), 695 (w), 680 (w), 575 (w).

#### 3.3.4. [Hg_2_Cl_4_(**L^2^**)_2_], **4**

Complex **4** was prepared by using similar procedures for **1**, except that HgCl_2_ (0.027 g, 0.10 mmol), **L^2^** (0.043 g, 0.10 mmol), and 10 mL MeOH were used. Colorless crystals were obtained. Yield: 0.053 g (76%). Anal. Calcd for C_48_H_40_Cl_4_Hg_2_N_8_O_8_ (*M*_W_ = 1399.86): C, 41.14; H, 2.86; N, 8.00%. Found: C, 40.75; H, 2.89; N, 7.91%. IR (cm^−1^): 3074 (w), 2939 (w), 2361 (w), 1768 (m), 1703 (s), 1612 (m), 1565 (w), 1429 (m), 1400 (m), 1347 (m), 1318 (m), 1220 (w), 1171 (m), 1108 (w), 1010 (w), 914 (m), 877 (w), 800 (w), 771 (m), 731 (w), 667 (w), 633 (m), 585 (w), 492 (m).

#### 3.3.5. [Hg_2_Br_4_(**L^2^**)_2_], **5**

Complex **5** was prepared by following the similar procedures for **4**, except that HgBr_2_ (0.036 g, 0.10 mmol) was used. Colorless crystals were obtained. Yield: 0.055 g (70%). Anal. Calcd for C_48_H_40_Br_4_Hg_2_N_8_O_8_ (*M*_W_ = 1577.66): C, 36.55; H, 2.54; N, 7.11%. Found: C, 36.69; H, 2.39; N, 7.08%. IR (cm^−1^): 3072 (w), 2933 (w), 2361 (w), 1768 (w), 1702 (s), 1611 (m), 1565 (w), 1429 (m), 1399 (m),1346 (m), 1317 (m), 1219 (w), 1170 (m), 1107 (w), 1011 (w), 915 (m), 876 (w), 798 (w), 770 (m), 731 (w), 667 (w), 634 (m), 586 (w), 492 (m).

#### 3.3.6. [Hg_2_I_4_(**L^2^**)_2_], **6**

Complex **6** was prepared by following the similar procedures for **4**, except that HgI_2_ (0.045 g, 0.10 mmol) was used. Colorless crystals were obtained. Yield: 0.039 g (44%). Anal. Calcd for C_48_H_40_Hg_2_I_4_N_8_O_8_ (*M*_W_ = 1765.66): C, 32.58; H, 2.26; N, 6.33%. Found: C, 32.72; H, 2.16; N, 6.31%. IR (cm^−1^): 3736 (w), 3567 (w), 3067 (w), 2967 (w), 2927 (w), 2362 (m), 2340 (w), 1942 (w), 1768 (m), 1701 (s), 1611 (m), 1565 (w), 1427 (m), 1396 (s), 1344 (m), 1316 (m), 1217 (m), 1169 (m), 1070 (m) 1009 (m), 914 (m), 875 (m), 796 (m), 769 (m), 729 (m), 696 (w), 667 (m).

#### 3.3.7. {[HgCl_2_(**L^3^**)]⋅H_2_O}_n_, **7**

Complex **7** was prepared by using similar procedures for **1**, except HgCl_2_ (0.028 g, 0.10 mmol), **L^3^** (0.043 g. 0.10 mmol) in 8 mL EtOH, and 2 mL H_2_O were used. Colorless crystals were obtained. Yield: 0.057 g (81%). Anal. Calcd for C_24_H_20_Cl_2_HgN_4_O_4_ (*M*_W_ = 699.93): C, 41.18; H, 2.88; N, 8.00%. Found: C, 41.02; H, 2.28; N, 7.56%. IR (cm^−1^): 3292 (s), 3046 (w), 1920 (m), 1660 (s), 1535 (s), 1505 (s), 1496 (s), 1414 (m), 1231 (s), 1170 (m), 1116 (s), 1097 (m), 1051 (w), 944 (m), 863 (m), 843 (m), 803 (m), 757 (m), 698 (m), 643 (w), 590 (m), 518 (w), 499 (w).

#### 3.3.8. {[HgBr_2_(**L^3^**)]⋅H_2_O}_n_, **8**

Complex **8** was prepared by following the similar procedures for **7**, except HgBr_2_ (0.036 g, 0.10 mmol) was used. Colorless crystals were obtained. Yield: 0.058 g (74%). Anal. Calcd for C_24_H_20_Br_2_HgN_4_O_4_ (*M*_W_ = 788.85): C, 36.54; H, 2.55; N, 7.10%. Found: C, 36.49; H, 2.88; N, 7.04%. IR (cm^−1^): 3280 (s), 3046 (w), 1914 (m), 1656 (s), 1539 (s), 1498 (s), 1479 (s), 1416 (m), 1233 (s), 1174 (m), 1109 (s), 1051 (m), 937 (m), 869 (m), 844 (m), 804 (m), 758 (m), 694 (m), 638 (w), 592 (m), 532 (w), 493 (w).

#### 3.3.9. {[HgI_2_(**L^3^**)]⋅H_2_O}_n_, **9**

Complex **9** was prepared by following the similar procedures for **7**, except HgI_2_ (0.045 g, 0.10 mmol) was used. Colorless crystals were obtained. Yield: 0.046 g (52%). Anal. Calcd for C_24_H_20_HgI_2_N_4_O_4_ (*M*_W_ = 882.83): C, 32.65; H, 2.28; N, 6.34%. Found: C, 33.18; H, 1.99; N, 6.06%. IR (cm^−1^): 3297 (s), 3057 (w), 1920 (m), 1653 (s), 1544 (s), 1496 (s), 1480 (s), 1414 (m), 1228 (s), 1171 (m), 1115 (s), 1052 (m), 1048 (w), 942 (m), 866 (m), 842 (m), 803 (m), 759 (m), 697 (m), 636 (w), 585 (m), 535 (w), 499 (w).

### 3.4. Powder X-ray Analysis

In order to check the phase purity of the product, powder X-ray diffraction (PXRD) experiments were carried out for complexes **1**–**9**. As shown in Appendix A, the peak positions of the experimental and simulated PXRD patterns were in good agreement with each other, indicating the bulk purities.

### 3.5. X-ray Crystallography

Single-crystal X-ray diffraction data for complexes **1**–**9** were collected on a Bruker AXS SMART APEX II CCD diffractometer with graphite-monochromated MoKα (λ_α_ = 0.71073 Å) radiation at 296 K [30]. Data reduction and absorption correction were performed by using standard methods with well-established computational procedures [31]. Some of the heavier atoms were located by the direct or Patterson method, and the remaining atoms were found in a series of Fourier maps and least-squares refinements, while the hydrogen atoms were added by using the HADD command in SHELXTL. Basic information pertaining to crystal parameters and structure refinement is listed in Table 6.

## 4. Conclusions

Nine Hg(II) halide complexes containing bpba ligands with bulky and angular spacers were successfully synthesized. Complexes **1**–**3** containing the rigid **L^1^** ligands with bulky backbones showed bizarre supramolecular structures that were dependent on the identity of the halide anions, whereas the structural types of **4**–**6** containing the flexible **L^2^** ligands with bulky spacer and **7**–**9** constructed from the angular **L^3^** ligands showed minimal dependence on the halide anions. The structural diversity of the bpba-based Hg(II) halide complexes and the effect of the halide anion are thus subject to the identities of the bpba ligands. Moreover, the bulkiness and the flexibility of the bpba ligands may also determine the effect of the halide anion. To further investigate the effect of halide anion on the structural diversity of flexible bpba-based Hg(II) complexes, future works can be geared towards the preparation of bpba ligands with more methylene groups [-(CH_2_)_n_-] in the backbone that can link the amide groups. Furthermore, reactions of mercury halide salts with flexible bpba ligands with C3 and C4 symmetries may also be investigated to afford Hg(II) complexes with interesting structural topologies. The sensing properties of **7**–**8** provide a unique insight into understanding the role of the halide anion in determining the quenching effect of the Hg(II) CPs by the metal ions, and the Hg-X---Fe^3+^ (X = Cl and Br) interactions may govern the quenching efficiency. Although toxic, Hg(II) halide complexes provide opportunities for the investigation of metal sensing.

## Figures and Tables

**Figure 1 ijms-23-07861-f001:**
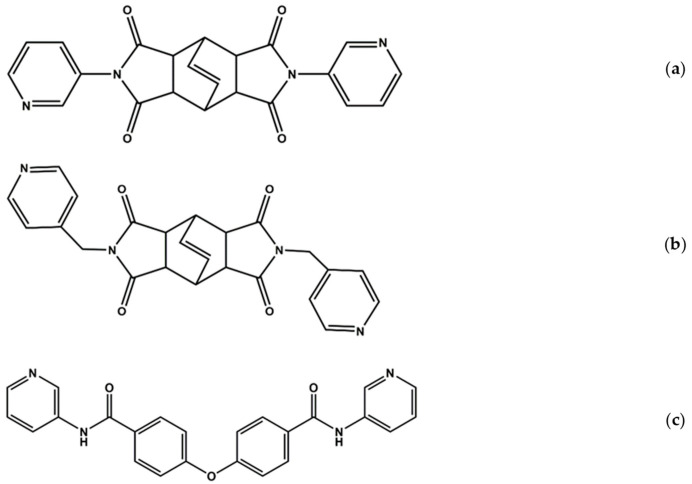
Structures of (**a**) **L^1^**, (**b**) **L^2^**, and (**c**) **L^3^**.

**Figure 2 ijms-23-07861-f002:**
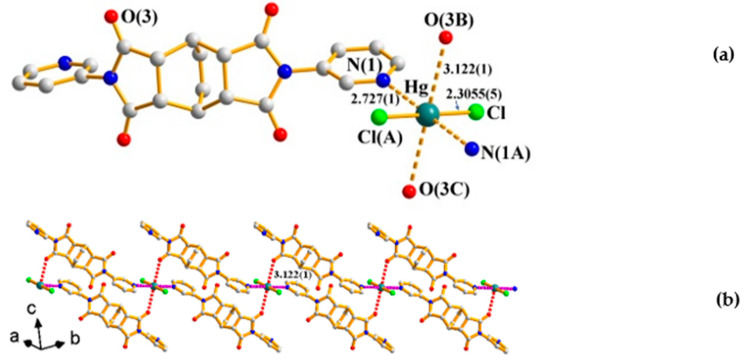
(**a**) Coordination environment of Hg(II) ion in **1**. Symmetry transformations used to generate equivalent atoms: (A) −x, −y + 2, −z + 1; (B) −x + 1, −y + 1, −z + 1; (C) x − 1, y + 1, z. (**b**) A depiction showing the 1D supramolecular structure of **1**.

**Figure 3 ijms-23-07861-f003:**
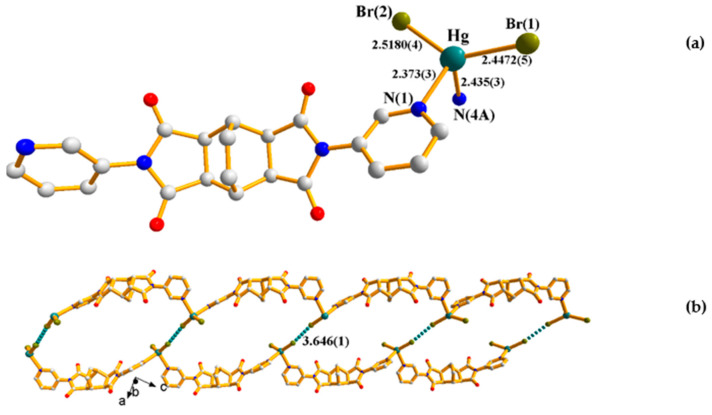
(**a**) Coordination environment about the Hg(II) ion in **2**. Symmetry transformations used to generate equivalent atoms: (A) x − 1, y, z + 1. (**b**) A drawing showing the 1D looped supramolecular chain.

**Figure 4 ijms-23-07861-f004:**
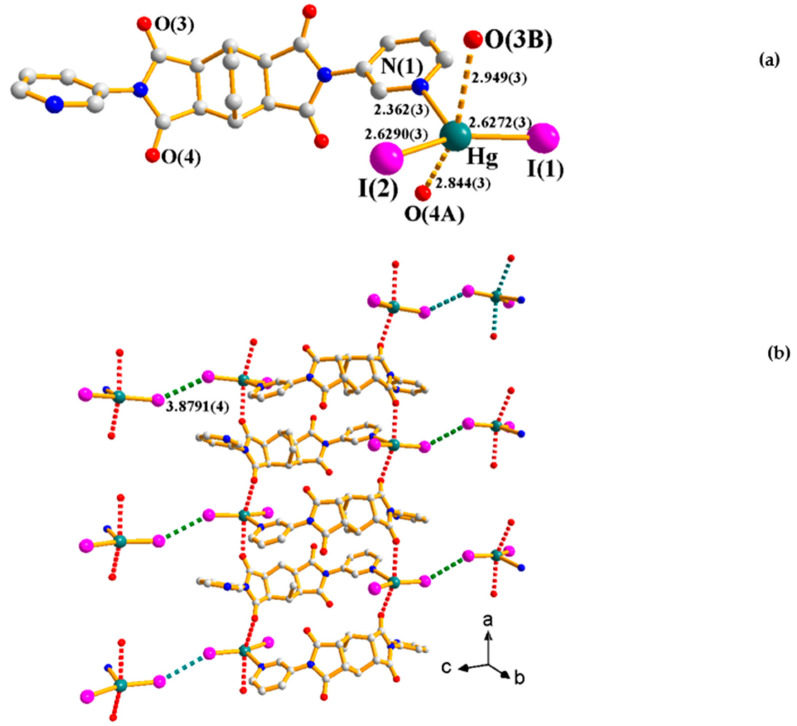
(**a**) Coordination environment of Hg(II) ion in **3**. Symmetry transformations used to generate equivalent atoms: (A) −x + 1, −y + 1, −z + 2; (B) −x + 2, −y + 1, −z + 2. (**b**) A drawing showing the 2D supramolecular layer.

**Figure 5 ijms-23-07861-f005:**
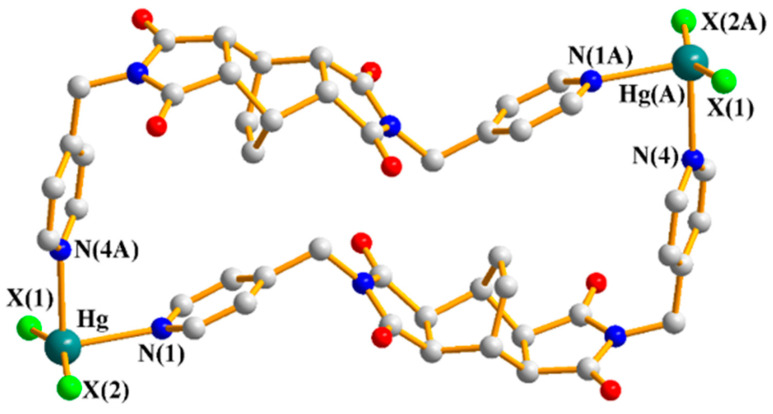
A representative drawing showing the dinuclear structures of **4** (X = Cl), **5** (X = Br), and **6** (X = I). Symmetry transformations used to generate equivalent atoms: (A) −x + 1/2, −y + 1/2, −z + 1 for **4**, −x + 1/2, −y + 3/2, −z + 1 for **5** and **6**.

**Figure 6 ijms-23-07861-f006:**
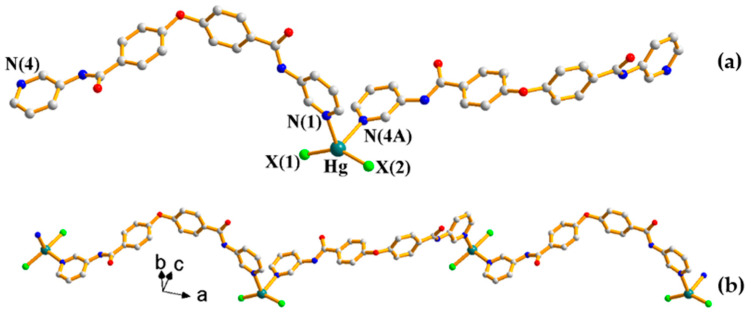
(**a**) A representative drawing showing the coordination environment about the Hg(II) ion for **7**–**9**. Symmetry transformations used to generate equivalent atoms: (A) x + 1, −y + 3/2, z + 1/2. (**b**) A drawing showing the 1D linear chain.

**Figure 7 ijms-23-07861-f007:**
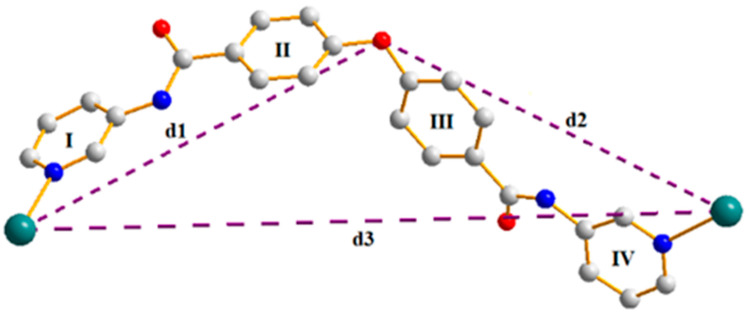
A schematic diagram defining the distances and the dihedral angle.

**Figure 8 ijms-23-07861-f008:**
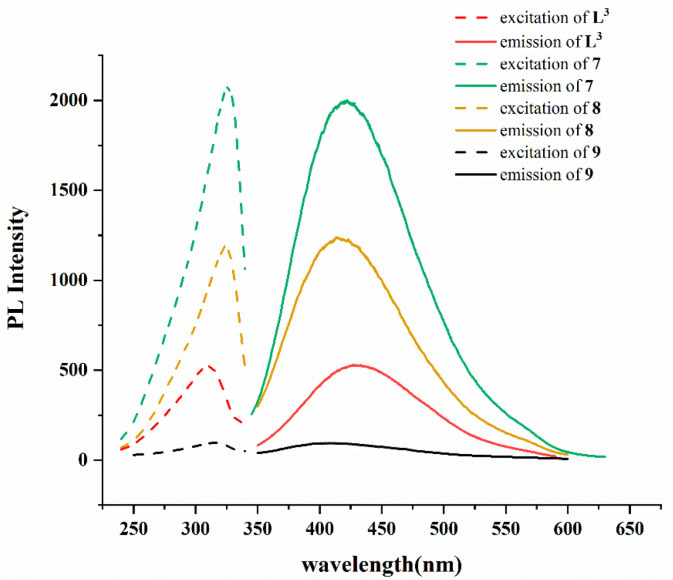
Excitation and emission spectra of complexes **7–9** and **L^3^**.

**Figure 9 ijms-23-07861-f009:**
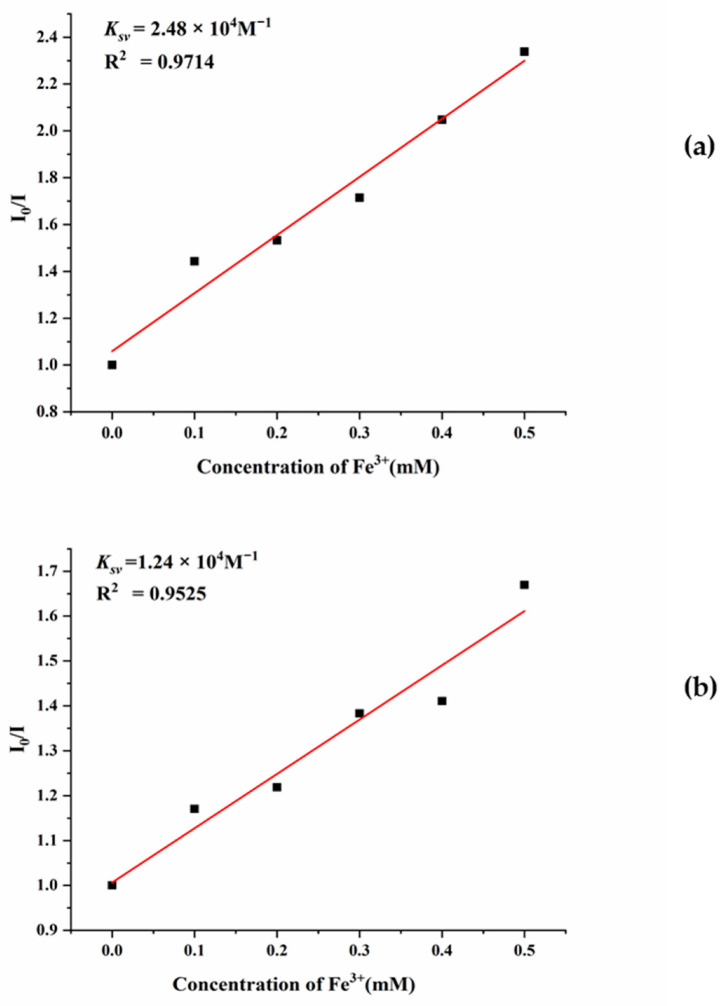
The Stern–Volmer plot of I_0_/I versus Fe^3+^ ions’ concentration for (**a**) **7** (λ_ex_ = 325 nm) and (**b**) **8** (λ_ex_ = 326 nm).

**Figure 10 ijms-23-07861-f010:**
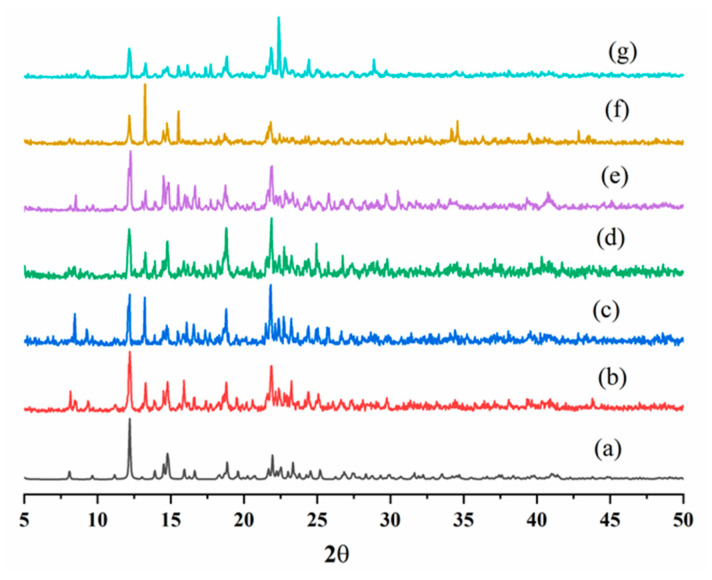
PXRD patterns before and after treatments with Fe^3+^ ions for **7**. (**a**) Simulated, (**b**) experimental, (**c**) 1st cycle, (**d**) 2nd cycle, (**e**) 3rd cycle, (**f**) 4th cycle, and (**g**) 5th cycle.

**Figure 11 ijms-23-07861-f011:**
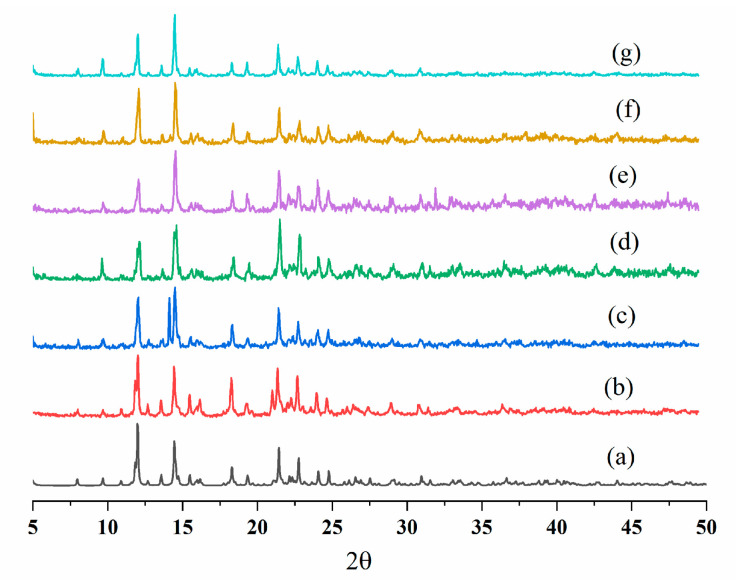
PXRD patterns before and after treatments with Fe^3+^ ions for **8**. (**a**) Simulated, (**b**) experimental, (**c**) 1st cycle, (**d**) 2nd cycle, (**e**) 3rd cycle, (**f**) 4th cycle, and (**g**) 5th cycle.

**Table 1 ijms-23-07861-t001:** Selected bond lengths (Å) and angles (°) for complexes **4** (X = Cl), **5** (X = Br), and **6** (X = I).

	4	5	6
Hg-N(1)	2.452(4)	2.503(4)	2.507(3)
Hg-N(4A)	2.510(4)	2.453(4)	2.459(3)
Hg-X(1)	2.3554(14)	2.4681(8)	2.6386(4)
Hg-X(2)	2.3461(15)	2.4670(8)	2.6322(4)
∠N(1)-Hg-N(4A)	84.27(13)	84.31(15)	83.92(12)
∠N(1)-Hg-X(1)	100.98(11)	98.54(10)	99.37(8)
∠N(1)-Hg-X(2)	97.22(10)	97.14(11)	100.10(9)
∠N(4A)-Hg-X(1)	97.47(10)	101.43(12)	102.67(9)
∠N(4A)-Hg-X(2)	94.91(10)	99.21(12)	100.18(9)
∠X(1)-Hg-X(2)	158.88(5)	155.15(2)	151.300(12)

**Table 2 ijms-23-07861-t002:** Selected bond distance (Å) and bond angles (°) for **7** (X = Cl), **8** (X = Br), and **9** (X = I).

	7	8	9
Hg-X(1)	2.3374(11)	2.4771(13)	2.6270(6)
Hg-X(2)	2.3858(10)	2.5213(12)	2.6732(6)
Hg-N(4A)	2.413(3)	2.419(9)	2.435(5)
Hg-N(1)	2.418(3)	2.388(8)	2.405(5)
∠X(1)-Hg-X(2)	147.79(4)	143.29(5)	139.976(18)
∠X(1)-Hg-N(4A)	105.38(8)	106.6(2)	107.22(12)
∠X(2)-Hg-N(4A)	97.52(7)	99.1(2)	100.84(12)
∠X(1)-Hg-N(1)	104.26(7)	106.4(2)	106.63(11)
∠X(2)-Hg-N(1)	99.16(7)	100.2(2)	102.16(11)
∠N(4A)-Hg-N(1)	87.10(10)	88.0(3)	88.18(18)

**Table 3 ijms-23-07861-t003:** Comparison of angles and distances for complexes **7**–**9**.

	d1 (Å)	d2 (Å)	d3 (Å)	θ1 (°)	θ2 (°)	θ3 (°)	C-O-C
**7**	11.05	11.50	19.80	21.75	77.79	2.11	118.3(2)
**8**	11.06	11.47	19.60	19.52	81.09	3.21	117.3(7)
**9**	11.06	11.40	19.41	16.89	84.51	5.94	117.3(4)

**Table 4 ijms-23-07861-t004:** Structural types of **1**–**9**.

Complex	Structure
[HgCl_2_]⋅2**L^1^_,_ 1**	1D supramolecular chain
[HgBr_2_(**L^1^**)]_n_, **2**	1D zigzag chain
[HgI_2_(**L^1^**)], **3**	2D supramolecular layer
[Hg_2_Cl_4_(**L^2^**)_2_], **4**	Dinuclear metallocycle
[Hg_2_Br_4_(**L^2^**)_2_], **5**	Dinuclear metallocycle
[Hg_2_I_4_(**L^2^**)_2_], **6**	Dinuclear metallocycle
{[HgCl_2_(**L^3^**)] H_2_O}_n_, **7**	1D zigzag chain
{[HgBr_2_(**L^3^**)] H_2_O}_n_, **8**	1D zigzag chain
{[HgI_2_(**L^3^**)] H_2_O}_n_, **9**	1D zigzag chain

**Table 5 ijms-23-07861-t005:** Luminescent properties of **7**–**9**.

Compound	Excitation λ_ex_ (nm)	Emission λ_em_ (nm)
**L^3^**	330	430
**7**	325	420
**8**	326	416
**9**	316	400

**Table 6 ijms-23-07861-t006:** Crystallographic data for **1**–**9**.

Compound	1	2	3
Formula	C_44_H_32_Cl_2_HgN_8_O_8_	C_22_H_16_Br_2_HgN_4_O_4_	C_22_H_16_HgI_2_N_4_O_4_
Formula weight	1072.26	760.80	854.78
Crystal system	Triclinic	Triclinic	Triclinic
Space group	*P*ī	*P*ī	*P*ī
a, Å	8.7814(14)	6.80779(8)	9.7275(5)
b, Å	10.5498(16)	11.83139(14)	11.5101(8)
c, Å	10.9631(16)	14.71857(16)	12.4521(7)
α, °	85.121(4)	80.4699(6)	106.3362(18)
*β*, °	74.166(4)	85.6414(6)	98.0437(13)
γ, °	81.889(4)	81.8897(6)	114.9454(11)
V, Å^3^	966.2(3)	1155.78(2)	1158.30(12)
Z	1	2	2
D_calc_, Mg/m^3^	1.843	2.186	2.451
F(000)	530	716	788
µ (Mo K_α_), mm^−1^	4.192	10.153	9.347
Range (2θ) for data collection, deg	3.86 ≤ 2θ ≤ 56.73	3.52 ≤ 2θ ≤ 56.60	3.57 ≤ 2θ ≤ 52.00
Independent reflections	4816 [R(int) = 0.0350]	5727 [R(int) = 0.0298]	4540 [R(int) = 0.0317]
Data/restraints/parameters	4816/0/286	5727/0/298	4540/0/298
quality-of-fit indicator ^c^	1.088	1.070	1.029
Final R indices [I > 2σ(I)] ^a,b^	R1 = 0.0142, wR2 = 0.0368	R1 = 0.0289, wR2 = 0. 696	R1 = 0.0158, wR2 = 0.0396
R indices (all data)	R1 = 0.0142, wR2 = 0.0368	R1 = 0.0327, wR2 = 0.0714	R1 = 0.0161, wR2 = 0.0397
**Compound**	**4**	**5**	**6**
Formula	C_48_H_40_Cl_4_Hg_2_N_8_O_8_	C_48_H_40_Br_4_Hg_2_N_8_O_8_	C_48_H_40_Hg_2_I_4_N_8_O_8_
Formula weight	1399.86	1577.70	1765.66
Crystal system	Monoclinic	Monoclinic	Monoclinic
Space group	*C*2*/c*	*C*2*/c*	*C*2*/c*
a, Å	25.8153(6)	26.208(2)	26.7552(5)
b, Å	7.0635(2)	7.1272(7)	7.2351(1)
c, Å	27.1652(7)	27.046(3)	27.0340(5)
α, °	90	90	90
*β*, °	97.5487(12)	97.866(5)	98.0783(9)
γ, °	90	90	90
V, Å^3^	4910.5(2)	5004.3(8)	5181.22(15)
Z	4	4	4
D_calc_, Mg/m^3^	1.893	2.094	2.264
F(000)	2704	2992	3280
µ (Mo K_α_), mm^−1^	6.525	9.383	8.362
Range (2θ) for data collection, deg	3.024 ≤ 2θ ≤ 56.656	3.04 ≤ 2θ ≤ 56.89	3.044 ≤ 2θ ≤ 56.678
Independent reflections	6126 [R(int) = 0.0372]	6265 [R(int) = 0.0588]	6439 [R(int) = 0.0371]
Data/restraints/parameters	6126/0/316	6265/0/298	6439/0/317
quality-of-fit indicator ^c^	1.059	1.025	1.033
Final R indices [I > 2σ(I)] ^a,b^	R1 = 0.0380, wR2 = 0.0951	R1 = 0.0397, wR2 = 0.0815	R1 = 0.0294, wR2 = 0.0674
R indices (all data)	R1 = 0.0506, wR2 = 0.1001	R1 = 0.0717, wR2 = 0.0904	R1 = 0.0363, wR2 = 0.0706
**Compound**	**7**	**8**	**9**
Formula	C_24_H_20_C_l2_HgN_4_O_4_	C_24_H_20_ Br_2_HgN_4_O_4_	C_24_H_20_HgI_2_N_4_O_4_
Formula weight	699.93	788.85	882.83
Crystal system	Monoclinic	Monoclinic	Monoclinic
Space group	*P*2**_1_**/c	*P*2_1_/c	*P*2_1_/c
a, Å	18.4292(3)	18.2680(16)	18.052(3)
b, Å	13.6689(2)	13.9475(11)	14.386(3)
c, Å	9.8057	9.9861(8)	10.1560(18)
α, °	90	90	90
*β*, °	90.3635(9)	91.326(4)	91.810(7)
γ, °	90	90	90
V, Å^3^	2470.09(7)	2543.7(4)	2636.3(8)
Z	4	4	4
D_calc_, Mg/m^3^	1.882	2.060	2.224
F(000)	1352	1496	1640
µ(Mo K_α_), mm^−1^	6.486	9.230	8.217
Range (2θ) for data collection, deg	3.71 ≤ 2θ ≤ 56.70	3.67 ≤ 2θ ≤ 52.11	3.62 ≤ 2θ ≤ 56.70
Independent reflections	6145 [R(Int) = 0.0357]	4984 [R(Int) = 0.0512]	6571 [R(Int) = 0.0448]
Data/restraints/parameters	6145/0/324	4984/0/321	6571/0/316
quality-of-fit indicator ^c^	1.048	1.044	1.034
Final R indices [I > 2σ(I)] ^a,b^	R1 = 0.0285, wR2 = 0.0689	R_1_ = 0.0535, wR_2_ = 0.1408	R_1_ = 0.0396, wR_2_ = 0.0929
R indices (all data)	R1 = 0.0366, wR2 = 0.0722	R1 = 0.0760, wR2 = 0.1506	R1 = 0.0592, wR2 = 0.1008

^a^ R_1_ = Σ||F_o_| − |F_c_||/Σ|F_o_|. ^b^ wR_2_ = [Σw(F_o_^2^ − F_c_^2^)^2^/Σw(F_o_^2^)^2^]^1/2^. w = 1/[σ^2^(F_o_^2^) + (ap)^2^ + (bp)], *p* = [max(F_o_^2^ or 0) + 2(F_c_^2^)]/3. a = 0.0141, b = 0.4522 for **1**; a = 0.0348, b = 1.304 for **2**; a = 0.0127, b = 3.7802 for **3**; a = 0.0453; b = 2.4854 for **4**; a = 0.0341 b = 5.0712 for **5**; a = 0.0296 b = 13.8466 for **6**; a = 0.0331, b = 2.1145 for **7**; a = 0.0712, b = 13.7591 for **8**; a = 0.0468, b = 4.1899 for **9**; ^c^ quality-of-fit = [Σw(|F_o_^2^| − |F_c_^2^|)^2^]/(N_observed_ − N_parameters_)^1/2^.

## Data Availability

Data are contained within the article or Appendix A.

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
