# Peer review of "Structural Diversity of Mercury(II) Halide Complexes Containing Bis-pyridyl-bis-amide with Bulky and Angular Backbones: Ligand Effect and Metal Sensing"

_ijms, 2022, doi:10.3390/ijms23147861_

Round 1

Reviewer 1 Report

The work presented by Chen et. al. is devoted to the rare field of Hg(II) coordination polymers. Examples of MOFs/complexes containing the used ligands are scarce. Structural features of compounds and luminescent properties have been studied. The sensing part of manuscript appears to be poor due to both its incompleteness and inapplicability of Hg(II) complexes the real analysis due to high toxicity of Hg(II). However, the paper is very well written and could be published in IJMS after the following improvements: 

1. UV/vis absorbance spectra do not prove the interaction between Fe3+ and the framework, as the luminescence decrease might be attributed just to the own absorbance of Fe3+ solution. Therefore, to determine a content of Fe3+ in its composites with 7-8 is crititally needed. EDX, atomic emission spectrocopy or any other type elemental analysis is recommended.

2. Fe3+ is much more acidic than other used cations, but their quenching is still quite high. To state the selective sensing properties of 7-8 to Fe3+, authors need to investigate a response to some cations having nature much more closer to Fe3+, e.g. Cr3+, Al3+, Zn2+, Pb2+

3. Uncoordinated water must be put outside of square brackets in formulas of 7-9 throughout the text. 

4. How the milling procedure has been performed - in a ball mill or manually? Please add this information to the experimental. 

5. UV/vis spestroscopy and luminescence instrumentation need to be described in the experimenal. Have the luminescent measurements been reproduced? What are the quantum yields in both solids and disperisons?

6. The information about purity of solvents needs to be added. 

Minor: 

7. Excitation wavelengths for Ksv determination graphs and for figs. 23-24 should be presented. 

8. A bulk formula of 9 might be corrected according to CHN data. 

9. A title in reference 19 should be corrected

Reviewer 2 Report

The authors provided a experimental study on structural diversities of 9 Bis-pyridyl-bis-amide containing mercury halide complexes. This is an interesting study, but cannot be published in its present form. My comments are as follows:

1.     The introduction part should be modified with the importance of the research field in more details, and why the authors choose these structures based on the previously published report.  There are several other reports also published. Those should also be taken into consideration.

2.     The figures 2, 3, 4 and 6 needs more clarity to understand the structures. Please add selected bond distances and angles in figures to enhance the quality of the figure.

3.     It is not clear why did the authors choose only structures 7-9 for metal sensing and ligand effect. They should make clear interpretation and comparison of all 9 structures.

4.     The authors stated they did not get any emission spectra for structure 1-6. Please clarify this part.

5.     Add selected XRD spectra in main manuscript.

6.     The conclusion part is too much general. There is no remark from authors and no future direction.

7.     The titles of subsections under Results and Discussion should be modified.

8.     There is no description about Figure 5.

9.     There are several unclear sentences and grammatical error. The ms should be revised by a native writer.  

Round 2

Reviewer 1 Report

Authors have addressed all the reviewers comments and now I can recommend the acceptance of this manuscript with great pleasure. 

Reviewer 2 Report

The authors of this study answered my comments and tried to modify their ms. But still, it does not ready for publication in its present form. I have a few comments to improve the quality of the ms.

1.     The authors must discuss some important similar works published by other research groups in Introduction section. The introduction part is too much general. It must be rewritten. The authors must discuss how and why their work is different/similar with other published report.

2.     The figures 2, 3 ,4, and 6 need more clarity. The authors just added bond distances in caption. This action of authors made the figures more complicated to understand. They must redraw the figures with some selected bond distances and angles.

3.     The conclusion section still lacks future direction.

4.     The ms still have some writing flaws. It must be checked by some native writer.

Round 3

Reviewer 2 Report

Authors of this work have considered my comments and have revised their paper. I can see the revision is feasible as red fonts in the revised ms. I think that this work is already improved compared to the original version of the ms. Thus, I suggest publication of this work